# Roles of Phenolic Compounds in the Reduction of Risk Factors of Cardiovascular Diseases

**DOI:** 10.3390/molecules24020366

**Published:** 2019-01-21

**Authors:** Mariane Lutz, Eduardo Fuentes, Felipe Ávila, Marcelo Alarcón, Iván Palomo

**Affiliations:** 1Thematic Task Force on Healthy Aging, CUECH Research Network, Universidad de Valparaíso, Angamos 655, Reñaca, Viña del Mar 2520000, Chile; edfuentes@utalca.cl (E.F.); favilac@utalca.cl (F.Á.); malarcon@utalca.cl (M.A.); ipalomo@utalca.cl (I.P.); 2Interdisciplinary Center for Health Studies, CIESAL, Faculty of Medicine, Universidad de Valparaíso, Angamos 655, Reñaca, Viña del Mar 2520000, Chile; 3Thrombosis Research Center, Department of Clinical Biochemistry and Immunohematology, Faculty of Health Sciences, Research Center for Aging, Universidad de Talca, 2 Norte 685, Talca 3460000, Chile; 4Escuela de Nutrición y Dietética, Facultad de Ciencias de la Salud, Universidad de Talca, Talca 3460000, Chile

**Keywords:** phenolic compounds, aging, cardiovascular disease, platelets, oxidative stress, atherothrombosis, age related diseases, health claims

## Abstract

The population is now living longer during the period classified as “elderly” (60 years and older), exhibiting multimorbidity associated to the lengthening of the average life span. The dietary intake of phenolic compounds (PC) may affect the physiology, disease development and progression during the aging process, reducing risk factors of age related diseases. The aim of this review is to briefly describe some of the possible effects of a series of PC on the reduction of risk factors of the onset of cardiovascular diseases, considering their potential mechanisms of action. The main actions described for PC are associated with reduced platelet activity, anti-inflammatory effects, and the protection from oxidation to reduce LDL and the generation of advanced glycation end products. Preclinical and clinical evidence of the physiological effects of various PC is presented, as well as the health claims approved by regulatory agencies.

## 1. Introduction

People are living longer. We are reaching old age in generally better health, and more individuals are now classified as “elderly” (e.g., 60 years and older), although a proportion exhibits multimorbidity [1]. Longevity is considered by the United Nations as one of the greatest achievements of our modern era, and a major social transformation of this century [2]. Current evidence strongly supports that diets rich in plant foods are associated with reduced risk of chronic diseases, including cardiovascular disease (CVD) [3]. In fact, some dietary priorities have been established in order to promote a general healthy aging, including the intake of minimally processed foods, bioactive rich foods (fruits, nuts, seeds, beans, vegetables, whole grains, plant oils, yogurt, and fish) and the reduction of the intake of refined starch, sugars, trans fats and sodium [4]. Accordingly, the current nutritional guidelines for the prevention of CVD include a diet high in fruits, vegetables and whole grains, nuts and legumes, and non-tropical vegetable oils, which is based on the Mediterranean (Med) diet, Dietary Approach to Stop Hypertension (DASH), AHA, and U.S. Department of Agriculture (USDA) dietary plans [5]. The protective mechanisms of these foods may not only be dependent on their nutrients and bioactive compounds content (e.g., antioxidants, anti-inflammatory agents), but also include their food matrix properties affecting glycemic load and energy density, among others. There has been a remarkable amount of research into health promoting constituents in foods, and this review focuses on various phenolic compounds (PC), molecules that derive from phenylalanine through pentose, shikimate, and phenylpropanoid phosphate biosynthetic pathways in plants, which participate in several important functions [6]. PC constitute a very diverse class of thousands of chemicals ubiquitously present in plants. There is emerging understanding of the underlying mechanisms of action of these compounds, and how they may affect the physiology, disease development and progression during the aging process [7]. Various PC may exert beneficial effects mainly by reducing risk factors of the onset of CVD, an age related disease (ARD) that involves “inflamm-aging” [8,9,10] since an up-regulation of inflammatory responses induces senescence and inflammatory changes [11]. Consequently, the aim of this review is to briefly describe some of the possible effects of the dietary intake of PC on the reduction of risk factors of the onset of CVD (a characteristic ARD), and their possible mechanisms of action. The focus is placed on antiinflammatory, antiplatelet aggregation, antioxidative, and antiglycating actions of PC which may contribute to a healthy aging process.

## 2. Dietary PC and Healthy Aging

Food intake plays a key role in modulation of the risk of cardiometabolic diseases. Over 30% of all deaths could be prevented through dietary changes, particularly by increased consumption of plant based foods [12]. The metabolic effects of the Med diet include lipid-lowering; as well as protection against oxidative stress, inflammation, and platelet aggregation; modification of hormones and growth factors involved in the pathogenesis of cancer; and gut microbiota-mediated production of metabolites influencing metabolic health [13], among others. The HALE cohort study was one of the first observational studies showing that individuals aged 70 to 90 years old adhering to a Med diet had 23% less overall 10 years mortality risk [14]. Various randomized clinical trials (RCTs) and epidemiological studies have demonstrated the importance of the Med diet [15] or dietary intake of vegetables, fruits, nuts, wholegrain cereals, fish, and olive oil in lowering inflammation and/or oxidative stress, which are risk factors CVD [16,17]. Such diets provide numerous bioactive compounds that improve health and well-being at every stage of life. Numerous studies have tried to identify the dietary constituents that exert protective effects on CVD, and dietary fiber, some fatty acids, and PC have been proposed [18,19]. In fact, dietary PC have a well-recognized role in reducing the risk of chronic diseases, the increase of healthy life years and the promotion of active healthy aging. Given the increase in CVD during aging, demand has raised for food constituents that may contribute to the prevention of this ARD and a better quality of people′s health [20]. 

The aging process involves the accumulation of diverse detrimental modifications in cells and tissues mainly due to oxidative changes that increase the risk factors of various ARDs and death. A high intake of dietary PC could potentially reduce the onset of ARDs, since they inhibit inflammatory mediators produced by cyclooxygenases (COX) COX-1 and COX-2 [21]. Consumption of PC may limit the incidence of coronary heart diseases, such as atherosclerosis, e.g., acting as potent inhibitors of LDL oxidation, considered to be a key mechanism in development of atherosclerosis [22]. Also, specific PC such as resveratrol has been described as an anti-aging agent, by extending the life span in various models, including the yeast *S. cerevisiae*, the fruit fly *Drosophila melanogaster*, the nematode *C. elegans* and the fish *Nothobranchius furzeri* [23]. Resveratrol, as well as caloric restriction, affect SIRT-1, which exerts beneficial effects on health and longevity [24]. It can also modify the action of forkhead box O (FOXO), proteins that regulate the expression of genes that contribute both to longevity and resistance to various stress mediators [25]. In humans, observational studies and RCTs correlate a long-term intake of PC with protection against ARDs, although the evidence is still controversial to some extent [26,27]. 

## 3. Bioavailability and Gut Microbiota Metabolism of PC

Following the ingestion of PC, only a selected amount of these components are absorbed into the circulatory system via the small intestine [28]. The absorptive process is dependent on their size, molecular complexity, charge, the food matrix, and the presence of other PC or drugs [29]. The upper gastrointestinal absorption of PC is low, and most part of the ingested PC pass into the colon [30]. Consequently, their bioavailability is low, and a substantial amount of the ingested phenolics is metabolized by the gut microbiota, generating small size molecules that may be absorbed and exert physiological effects [31,32,33]. Thus, the level of urinary excretion of PC indicates that the colonic catabolites are absorbed into the portal vein and circulate through the body prior to excretion [34]. Gut metabolites are able to catabolize even the non-extractable PC, releasing small molecules that are efficiently absorbed and may exert anti-inflammatory and antioxidant effects. A classic example of this is the effect of the microbiota metabolites of urolithins, derived from ellagitannins (abundant in pomegranate), which can exert their bioactivity for longer periods than the original PC absorbed directly in the small intestine [35]. Thus, the urine metabolites profile may be markedly different to that of plasma [36]. Additionally, since PC may positively modulate the gut microbiota composition and function, acting as prebiotics, these compounds may also indirectly lower the risk of CVD [37]. Moreover, the simultaneous intake of various plant sources providing a variety of PC results in changes in their bioavailability. Food combinations provide synergistic effects on inhibiting oxidation and inflammation, among other physiological activities, depending on the bioaccessibility (fraction released from the food matrix at the gastrointestinal tract, available for absorption) and the bioavailability of the PC ingested, which is affected by the presence of other phytochemicals [38]. On the other hand, the implications of food processing on the bioavailability of PC should also be taken into consideration [39]. 

## 4. PC and CVD

Nearly half of non-communicable diseases are represented by CVD [40]. Globally, the number of deaths due to CVD between 1990 and 2013 increased by 41%, climbing from 12.3 to 17.3 million [41,42]. Risk factors such as age, high cholesterol levels, high blood pressure, diabetes, urinary isoprostanes, LDL oxidation, platelet aggregation, and inflammatory status, among others, exert an effect through deregulated endothelium and hyperactivated platelets [43,44,45,46], contributing to atherothrombosis and the development of CVD [47,48]. Platelet activation and subsequent accumulation at sites of vascular injury play central roles in thrombus formation, which is believed to be the trigger of CVD, such as atherosclerosis, myocardial infarction and stroke [49]. 

Various studies have demonstrated the CVD protective role of PC, acting on the prevention of atherothrombosis, mainly based on the in vitro antiplatelet activity of a number of fruits and vegetable extracts [50,51]. Most described roles of dietary PC relate with their ability to synergize affecting endothelial damage, platelets reactivity, and oxidative damage, among other processes that are relevant in the development of ARDs and an unhealthy aging. Some described effects include the contribution of PC to stabilize the atheroma plate, e.g., quercetin inhibits the action of metalloproteinase I (MMP-1) and prevents the disruption of plaques [52]; Chinese tea catechins inhibit the invasion and proliferation of smooth muscle cells of the arterial wall, reducing the formation of the atheromatose lesion [53]; PC in general reduce platelet aggregation [54]; while the PC contained in olive oil decrease the oxidation of LDL and triglyceride levels, and increase HDL levels in rats [55,56].

## 5. Anti-Inflammatory and Anti-Platelet Aggregation Effects of PC

Primary and secondary prevention of atherothrombosis is largely accomplished by using pharmacological agents such as aspirin and glycoprotein inhibitors [57]. However, middle aged and older individuals, smokers and people under stress can also display impaired platelet and endothelial activities. Therefore, a healthy dietary approach to maintaining CV health may be considered another important tool in the prevention of atherothrombosis [58]. PC may be considered as natural inhibitors of platelet aggregation, helping to reduce the individual risk of developing CVD involving thrombosis, and an expedient alternative to a pharmacological approach to the protection of the population at risk (Table 1).

Arachidonic acid (ARA, 20:4n-6) metabolism by COX is a major pathway for blood platelets′ activation, which is associated with pro-inflammatory prostaglandins (PGs) and pro-thrombotic thromboxane A_2_ (TXA_2_), a very strong blood platelet activator via thromboxane receptor acting as a pro-aggregator and vasoconstrictor mediator, leading to increased platelet aggregation [68,69]. Inhibition of COX activity is one of the major means of anti-platelet pharmacotherapy (acetylsalicylic acid) by reducing the production of TxA_2_ and subsequently inhibiting platelet aggregation and formation of the platelet plug [70]. In vitro studies have described a variety of PC exhibiting a significant antiplatelet potential via inhibition of platelet ARA pathway. Among these, Mattiello et al. [71] reported an antiplatelet effect of PC-rich extract from pomegranate (POMx), able to reduce platelet aggregation at low concentrations of phenols. POMx inhibited ARA and collagen-induced platelet aggregation with maximal percentage of platelet aggregation. Such is the case for PC like 8-paradol, geranylgeraniol, genistein, daidzein, silychristin, silybin, ginkgetin, apigenin, cycloheterophyllin, broussoflavonol F, quercetin, and hesperetin, among others found in various plant foods (e.g., capers, blueberries, radish, coriander, oregano, onion, asparagus, apples, tomato, cranberries, lettuce, potato). These molecules possess a marked COX-1 inhibitory activity [72,73,74,75,76]. Nurtjahja-Tjendraputra et al. [72] determined the inhibitory activity of 8-paradol and gingerol synthetic analogues against COX-1. Interestingly, 8-paradol exhibited the strongest COX-1 inhibitory activity, with IC_50_ values of 4 ± 1 μM, while gingerol synthetic analogues exhibited lower potency (IC_50_ ~20 μM). Calixto et al. [73] showed that geranylgeraniol prevents TXB_2_ formation induced by ARA in a dose-dependent manner, and shows inhibition indexes of 28.7%, 54.6%, and 89.4% at 1, 10, and 100 nmol/mL, respectively. Geranylgeraniol also prevents PGE_2_ formation induced by ARA in a dose-depedent manner (1, 10, 100 nmol/mL), with inhibition indexes of 49.1%, 69.6%, and 89.7%, respectively. These results suggest that COX-1 is the primary target of this PC. Chien-Ming et al. [74] showed that ginkgetin, apigenin, cycloheterophyllin, broussoflavone F, and quercetin inhibited COX-1 activity. Ginkgetin was the most potent inhibitor with an IC_50_ of 100 μM. In crystallization experiments, all COX-1 inhibitors interacted with the putative catalytic aminoacid residue Tyr 385 and formed hydrogen bonds with Arg 120 and Tyr 355 [74]. When mapping these components, all of them docked near the gate of COX-1. It could be observed that the phenolic oxygen atom of flavonoids accepts hydrogen bond from Arg 120 or Tyr 355 [74]. Karlíčková et al. [75] showed that the isoflavones genistein and daidzein were more potent inhibitors of COX-1 activity at 100 μM. In case of daidzein, a threshold effect at ~40% was observed. Finally, Bijak et al. [76] showed that silybin, silychristin and silydianin also inhibit COX-1. The strongest inhibitory effect was observed in silychristin and silybin (IC_50_ 35 μM and 60 μM, respectively). In silico, both compounds interact with the active site of COX-1, blocking the possibility of substrate binding. Both PC interact with Tyr residues of the enzyme. They exhibit a strong binding mode to COX-1 active site entry (−9.8 kcal/mol and −9.2 kcal/mol respectively) compared to flurbiprofenum, one of the most popular non-steroid anti-inflammatory drugs (−8.9 kcal/mol) [77]. Phloroglucinol (naturally present in brown seaweed, however it is used mainly as a pure synthetic chemical) has shown antiplatelet effect by inhibiting COX-1 and COX-2 by 45–74% and 49–72%, respectively, at concentrations of 10–50 μM. Some findings have been observed in animal models, for instance the intravenous administration of phloroglucinol in mice (2.5 and 5 μmol) suppressed the ex vivo ARA-induced platelet aggregation by 57–71% [78].

Other PC act as antagonists on thromboxane receptors. For instance, the antiplatelet activity of tectorigenin (found in the lily *Belamcanda chinensis*) is not based on inhibition of COX-1 (in contrast to acetylsalicylic acid or thromboxane synthase), but acts as a competitive antagonist at thromboxane receptors [66]. Also, the antiplatelet effect of various PC such as apigenin (e.g., parsley, peppermint, thyme), luteolin (e.g., oregano, peppermint, radicchio), genistein (e.g., soybean), quercetin (e.g., capers, lovage, radish), and carnosol (e.g., rosemary) might in part rely on TxA2 receptor antagonism [79,80,81]. Tight binding of flavonoids to TxA_2_ receptor relies on structural features such as the presence of the double bond in C2-C3, and a keto group in C4 [80]. On the other hand, the production of the ARA metabolite 12-hydroxy-5,8,10,14-eicosatetraenoic acid (12-HETE), was significantly lower in ginsenoside Rk1 or ajoene-treated platelets compared to the ARA-induced group [82,83].

## 6. PC as Antioxidative Agents

Oxidative stress affects virtually all aspects of human health. This includes all organs and the system as a whole. Inflammation and oxidative stress are linked, and both situations increase with aging. The generation of reactive oxygen species (ROS) takes place in neutrophils and macrophages during inflammation and in other processes of normal cellular metabolic activities [84], and mitochondrial ROS contribute to inflammatory responses and have been implicated in the activation of major inflammatory transcription factors such as NF-kB, AP-1 and the inflammasome [85,86]. The imbalance between ROS production and the ROS-detoxifying systems can promote atherogenesis through a number of events including mitochondrial and DNA damage, endoplasmic stress and pro-inflammatory effects leading to endothelial cell activation, vascular smooth muscle cell proliferation and immune cell activation. A clear consequence of the deleterious effects of oxidative stress is the generation of oxidized LDL, that plays a crucial role in atherosclerosis initiation and progression, being a major constituent of atherogenic plaques, the basis of CVD [87,88]. PC have been considered as intracellular direct antioxidants, by scavenging oxidizing species, and to exert an indirect effect, by inducing the up-regulation of the synthesis of endogenous antioxidant enzymes such as superoxide dismutase (SOD), catalases, peroxidases such as selenium-dependent glutathione peroxidases (GPx), and paraoxonases, which degrade ROS like superoxide and hydroperoxides [89,90]. The protective effects of PC against various ARDs is well recognized, and animal and clinical studies have reported beneficial effects of their intake on a number of risk factors for CVD, including LDL cholesterol, blood pressure, and endothelial function [91,92,93].

Endothelial cells regulate vascular homeostasis by producing factors that act locally in the vessel wall and lumen, including nitric oxide (NO), which has important vasodilator, antiinflammatory, antithrombotic, and growth suppressing properties that are relevant to all stages of atherosclerosis [94]. Since the classical Zutphen Study was accomplished, it has been observed that the dietary intake of flavonoids correlated inversely with CVD [95], and in a sub-population of this study it was observed that cocoa consumption was associated with a decrease in blood pressure and overall CVD mortality [96,97]. In fact, some specific PC, such as cocoa flavanols, act on endothelial function [98,99,100], and several clinical studies have shown their beneficial effects. Among these, the FLAVIOLA Health Study (FHS) showed that the daily intake of cocoa flavanol supplementation for 1 month (450 mg) improved the Framingham Risk Score in relatively healthy men and women. In a revision of RCTs investigating the effects of chocolate or cocoa products on systolic and diastolic blood pressure in adults for a minimum of two weeks duration, Ried et al. [101] observed a moderate-quality evidence that flavanol-rich chocolate and cocoa products cause a small (2 mmHg) blood pressure-lowering effect in the short term, and signal that long-term trials investigating the effect of cocoa on clinical outcomes are needed to assess whether it has an effect on CV events and to assess potential adverse effects associated with chronic ingestion of cocoa products. 

## 7. PC as Antiglycating Agents

The accumulation of advanced glycation end products (AGEs) has been implied in the etiology of numerous non-communicable diseases in the aging process [102,103]. The accumulation of AGEs can take place by the intake of glycated dietary proteins or in situ reactions occurring as a result of the normal metabolism. RCTs and epidemiological studies have shown that decreasing the levels of dietary AGEs can have beneficial effects represented by a decrease in CV risk factors (total cholesterol and LDL), increase of protective systems (AGE-R1 and Glyoxalase I), and a decrease in markers of insulin resistance, oxidative stress and inflammation [104,105]. In most RCTs involving diets reduced in AGEs, the cooking methods avoided exposure to dry heat and favored the use of low temperatures and high-water content [106]. Rodriguez et al. [107] showed that the intake of Med diet (excluding wine intake) decreased the levels of serum carboxymethyl-lysine (CML). In agreement with this, it has been reported that Med diets without or with coenzyme Q10 supplementation is able to decrease the levels of serum CML after 4 h [108]. The thermal processing of turkey meat with an antioxidant rich concentrate containing native Chilean berries reduced the malondialdehyde (MDA) levels after cooking, possibly reducing the generation of lipoperoxidation end products [109]. The intake of a 250 g turkey burger with 500 mL of a berry concentrate drink (1.39 ± 0.05 mg GAE/g) by healthy volunteers reduced the postprandial MDA/triglycerides area under the curve by 35% compared with the response induced by a 250 g turkey burger with 500 mL of water [109]. Strikingly, when volunteers consumed a 250 g turkey burger cooked with PC (380 mg GAE) and 500 mL of a berry concentrate drink (1.39 ± 0.05 mg GAE/g), the rise in the levels of postprandial MDA/ triglycerides was inhibited. These results are relevant, since MDA is not only a product of polyunsaturated fatty acids oxidation, but it can also react with amino acids and proteins, generating adducts such as MDA-Lys which has been implied in the etiology of CVD [110] and shows an inverse correlation with the maximum lifespan in mammal tissues [111,112].

Besides their antioxidant effects, PC can also react with dicarbonyl and α-β unsaturated compounds, inhibiting the generation of AGEs in proteins. It has been determined that antiglycation activity of Canadian native berries and tropical medicinal herbs correlate strongly with the free radical scavenging activity and total PC content [113,114]. Accordingly, numerous PC extracts have been shown to prevent the accumulation of AGEs by both cellular responses and direct scavenging of reactive species [115]. The antiglycating mechanisms involve the activation of preventive mechanisms, such as the increase in glutathione synthesis and the overexpression of glyoxalase 1 [115,116], as well as an increase of the removal mechanisms of glycated proteins such as proteasome and cathepsins D and L, among others [117]. On the other hand, the scavenging analysis of the levels of methylglyoxal after incubation with PC has shown that the number of hydroxyl groups by molecule is a key factor in the reaction, the efficiency being monohydroxylated < dihydroxylated < trihydroxylated benzene ring [118]. Flavonoids such as quercetine, catechin, epicatechin and anthocyanidines react with methylglyoxal [119,120,121,122], and structural analysis of the reaction products has shown that the addition occurs in the A ring of the flavonoid. Stilbenes such as 2,3,5,4′-tetrahydroxystilbene-2-*O*-β-d-glucoside and resveratrol also show scavenging activity, inhibiting the glycation of BSA induced by fructose and methylglyoxal [123,124]. However, a double blind, placebo-controlled RCT with a crossover design demonstrated that after dietary supplementation of healthy volunteers during 4 weeks with equimolar amounts of epicatechin (100 mg/d), quercetin 3-glucoside (160 mg/d) or placebo, only quercetin decreased plasma methylglyoxal (10.6%) [125]. The same doses of epicatechin and quercetin 3-glucoside administered to healthy subjects resulted in a similar increase of these compounds in plasma (~2000 nmol/L) after 2 h of administration [126]. Therefore, the differences in the effects mediated by these flavonoids could be explained in part by the elimination half-life, which has been reported to be 17 h for quercetin and 2 h for epicatechin [125]. More studies are needed to establish the actual effect of antiglycating activity of PC, especially during the thermal treatment of foods, considering that oxidation products can result in quinones which can bind covalently to proteins, changing their nutritional quality and sensory properties [126,127].

## 8. Health Claims Regulation and Association with PC

In this brief review some of the potential physiological effects of dietary PC on the reduction of risk factors of ECV, assayed by in vitro, in vivo and RCTs are described. The substantiation of the beneficial effects of bioactive constituents of foods, such as PC, is a requisite for the approval of a health claim [128]. Preclinical, as well as observational and epidemiological studies are not considered as enough evidence to approve a health claim, and only well designed clinical intervention studies (mainly RCTs) are considered as the standard to provide evidence for the scientific substantiation of specific claims. Moreover, health claims substantiation demand to establish cause-effect relationships [129]. Some regulatory agencies, such as the European Food Safety Agency [130,131] and the Food and Drug Administration [132] assist applicants for the authorization of health claims related to the antioxidants, oxidative damage and CV health. A series of outcome variables have been proposed for the scientific substantiation of general claims on CV health, such as beneficial changes in the blood lipids levels (e.g., reduction in LDL for claims on maintenance of normal blood concentrations of LDL and/or triglycerides and/or an increase in blood HDL -as long as blood LDL are not increased-), arterial blood pressure, elastic properties of the arteries, endothelial function, plasma homocysteine concentrations, platelet aggregation and venous blood flow. All these variables can be assessed in clinical studies and allow the scientific evaluation of specific function claims in the area of CV health [133]. Several disease risk reduction claims related to CVD risk using elevated LDL as the risk factor biomarker have been approved. However, few health claims in association with ECV have been approved for PC. In fact, the EFSA has only approved health claims for olive oil hydroxytyrosol and cocoa flavanols, mainly due to the insufficient scientific evidence of the beneficial effects of PC obtained by clinical trials. 

## 9. Conclusions

The aging process involves a series of physiological challenges, which are significantly affected by dietary factors (nutrients and bioactive compounds). In this review, the results of a series of studies of the potentially beneficial action of ingested PC on some major selected risk factors of ARDs are described. While in vitro and in vivo studies (preclinical) are mainly related to the possible mechanisms of action of PC, RCTs represent a closer approach for the substantiation of the effects of these molecules in humans and allow the possibility to use a health claim for some foods containing them. This relevant information is basic when considering the association of PC with a healthy aging process, since the development of CVD may be retarded through the intake of a myriad of dietary PC affecting the risk factors involved.

## Figures and Tables

**Table 1 molecules-24-00366-t001:** PC affecting platelet aggregation induced by different agonists.

Phenolic Compound	Amount (μM)	Agonist	% Inhibition	Reference
Caffeic acid	478	Collagen (1 μg/mL)	50	[59]
	14	ADP (3 μM)	0	[60]
p-Coumaric acid	483	Collagen (1 μg/mL)	50	[59]
Ferulic acid	482	Collagen (1 μg/mL)	50	[59]
Gallic acid	100	Thrombin (0.1 UI/mL)	10	[61]
Quercetin	102	ADP (5 μM)	50	[62]
	55	Collagen (2.5 μg/mL)	50	[62]
	330	Thrombin (0.1 UI/mL)	50	[62]
	18	ARA (45 μg/mL)	50	[63]
Myricetin	420	ADP (4 μM)	0	[64]
	420	Collagen (1 μg/mL)	0	[64]
	100	Thrombin (1.5 UI/mL)	0	[64]
	55	ARA (45 μg/mL)	50	[63]
Catechin	420	ADP (10 μM)	0	[64]
	420	Collagen (1 μg/mL)	0	[64]
	420	Thrombin (1.5 UI/mL)	0	[64]
	420	ARA (500 μg/mL)	0	[64]
Epicatechin	100	Thrombin (0.1 UI/mL)	0	[64]
Apigenin	420	ADP (10 μM)	0	[64]
	420	Collagen (1 μg/mL)	0	[64]
	420	Thrombin (1.5 UI/mL)	0	[64]
	420	ARA (500 μg/mL)	0	[64]
Luteolin	100	Collagen (2 μg/mL)	23	[65]
	200	Thrombin (0.5 UI/mL)	50	[66]
	57	ARA (38 μg/mL)	50	[66]
Hesperetin	200	Collagen (5 μg/mL)	50	[63]
	120	ARA (45 μg/mL)	50	[63]
Genistein	12	Collagen (5 μg/mL)	50	[66]
	100	Thrombin (0.5 UI/mL)	50	[66]
	24	ARA (38 μg/mL)	50	[66]
Resveratrol	11	ADP (3 μM)	0	[60]
	50	Collagen (5 μg/mL)	0	[67]
	1000	Thrombin (0.12 UI/mL)	10	[68]

ADP: adenosine diphosphate; ARA: arachidonic acid.

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
