# Peer review of "Roles of Phenolic Compounds in the Reduction of Risk Factors of Cardiovascular Diseases"

_molecules, 2019, doi:10.3390/molecules24020366_

Round 1

Reviewer 1 Report

Reviewer comment: 

The manuscript entitled “Roles of phenolic compounds on the reduction of risk factors of cardiovascular diseases” focus on an interest theme for Molecules, although it needs to be improved in order to be considered for publication.

The main flaw of the manuscript is associated with the fact it presented in a highly subjective way. Hence, I suggest some major corrections:

a) In general, studies included in the distinct subtopics of the manuscript seem described in a non-detailed way and must be more detailed (just as an example, text in lines 163-169 should be detailed and main conditions/results of the works should be summarized in a Table).  

b) Authors should introduce Materials and methods section, elucidating which studies were included into review and why; the keywords used during literature search, inclusion and exclusion criteria of references, information about the procedure of literature search conducted by them, number of chosen references, as well as information if some of them were excluded from the review and on the basis of which criteria.

c) Lines 136- 139: “PC may be considered as natural inhibitors of platelet aggregation, helping to reduce the individual risk of developing CVD involving thrombosis, and an expedient alternative to a pharmacological approach to the protection of the population at risk (Table 1).”- Please detail further conditions/studies that support these conclusions.

d) strengths and limitations of this work must be highlighted in before or within the conclusion section (lines 701-707).

Others: 

Lines 57: change to “Food intake plays a key role in modulation of the risk of cardiometabolic…”

Rephrase lines 34-36 for clarification

Author Response

The manuscript focus on an interest theme for Molecules, although it needs to be improved in order to be considered for publication. The main flaw of the manuscript is associated with the fact it presented in a highly subjective way. Hence, I suggest some major corrections:

In general, studies included in the distinct subtopics of the manuscript seem described in a non-detailed way and must be more detailed (just as an example, text in lines 163-169 should be detailed and main conditions/results of the works should be summarized in a Table). 

Answer: A new paragraph was introduced (page 4) and Table 1 was modified summarizing the findings (page 10).

Modified text: PC inhibit platelet aggregation, contributing to reduce the individual risk of developing CVD involving thrombosis, through a series of mechanisms of action. Arachidonic acid (ARA, 20:4n-6) metabolism by COX is a major pathway for blood platelets' activation, which is associated with pro-inflammatory prostaglandins (PGs) and pro-thrombotic thromboxane A2 (TXA2), a very strong blood platelet activator via thromboxane receptor acting as a pro-aggregator and vasoconstrictor mediator, leading to increased platelet aggregation [59]. Inhibition of COX activity is one of the major means of anti-platelet pharmacotherapy (acetylsalicylic acid) by reducing the production of TxA2 and subsequently inhibiting platelet aggregation and formation of the platelet plug [60]. In vitro studies have described a variety of PC exhibiting a significant antiplatelet potential via inhibition of platelet ARA pathway. Among these, Mattiello et al. [61] reported an antiplatelet effect of PC-rich extract from pomegranate (POMx), able to reduce platelet aggregation at low concentrations of phenols. POMx inhibited ARA and collagen-induced platelet aggregation with maximal percentage of platelet aggregation. Such is the case for PC like 8-paradol, geranylgeraniol, genistein, daidzein, silychristin, silybin, ginkgetin, apigenin, cycloheterophyllin, broussoflavonol F, quercetin, and hesperetin, among others found in various plant foods (e.g. capers, blueberries, radish, coriander, oregano, onion, asparagus, apples, tomato, cranberries, lettuce, potato). These molecules possess a marked COX-1 inhibitory activity [62-66]. Nurtjahja-Tjendraputra et al. [62] determined the inhibitory activity of 8-paradol and gingerol synthetic analogues against COX-1. Interestingly, 8-paradol exhibited the strongest COX-1 inhibitory activity, with IC50 values of 4±1  mM, while gingerol synthetic analogues exhibited lower potency (IC50 ~ 20 mM). Oliveira et al. [63] showed that geranylgeraniol prevents TXB2 formation induced by ARA in a dose-dependent manner, and shows inhibition indexes of 28.7%, 54.6%, and 89.4% at 1, 10, and 100 nmol/mL, respectively. Geranylgeraniol also prevents PGE2 formation induced by ARA in a dose-depedent manner (1, 10, 100 nmol/mL), with inhibition indexes of 49.1%, 69.6%, and 89.7%, respectively. These results suggest that COX-1 is the primary target of this PC. Wu et al. [64] showed that ginkgetin, apigenin, cycloheterophyllin, broussoflavone F, and quercetin inhibited COX-1 activity. Ginkgetin was the most potent inhibitor with an IC50 of 100 mM. In crystallization experiments, all COX-1 inhibitors interacted with the putative catalytic aminoacid residue Tyr 385 and formed hydrogen bonds with Arg 120 and Tyr 355 [64]. When mapping these components, all of them docked near the gate of COX-1. It could be observed that the phenolic oxygen atom of flavonoids accepts hydrogen bond from Arg 120 or Tyr 355 [64]. Karlíčková et al. [65] showed that the isoflavones genistein and daidzein were more potent inhibitors of COX-1 activity at 100 μM. In case of daidzein, a threshold effect at ~ 40% was observed. Finally, Bijak et al. [66] showed that silybin, silychristin and silydianin also inhibit COX-1. The strongest inhibitory effect was observed in silychristin and silybin (IC50 35 μM and 60 μM, respectively). In silico, both compounds interact with the active site of COX-1, blocking the possibility of substrate binding. Both PC interact with Tyr residues of the enzyme. They exhibit a strong binding mode to COX-1 active site entry (−9.8 kcal/mol and −9.2 kcal/mol respectively) compared to flurbiprofenum, one of the most popular non-steroid anti-inflammatory drugs (−8.9 kcal/mol) [66]. Phloroglucinol (naturally present in brown seaweed, however it is used mainly as a pure synthetic chemical) has shown antiplatelet effect by inhibiting COX-1 and COX-2 by 45-74% and 49-72%, respectively, at concentrations of 10-50 μM. Some findings have been observed in animal models, for instance the intravenous administration of phloroglucinol in mice (2.5 and 5 μmol) suppressed the ex vivo ARA-induced platelet aggregation by 57-71% [67].

Authors should introduce Materials and methods section, elucidating which studies were included into review and why; the keywords used during literature search, inclusion and exclusion criteria of references, information about the procedure of literature search conducted by them, number of chosen references, as well as information if some of them were excluded from the review and on the basis of which criteria.

Answer: We did not introduce a Materials and Methods section, since the manuscript is not a systematic review and every author contributed with the information considered as relevant for each of the subjects covered, describing different aspects of the potential actions of phenolic compounds on the reduction of risk factors involved in the development of age related diseases (mainly CVD).

Lines 136- 139: “PC may be considered as natural inhibitors of platelet aggregation, helping to reduce the individual risk of developing CVD involving thrombosis, and an expedient alternative to a pharmacological approach to the protection of the population at risk (Table 1).”- Please detail further conditions/studies that support these conclusions.

Answer: The phrase was replaced by “PC inhibit platelet aggregation, contributing to reduce the individual risk of developing CVD involving thrombosis, through a series of mechanisms of action”. The paragraph was modified, including a more exhaustive description of studies of the various mechanisms of action of PC, and Table 1 was modified accordingly.

Strengths and limitations of this work must be highlighted in before or within the conclusion section (lines 701-707).

Answer: The conclusions were re-written as recommended.

Modified text: The aging process involves a series of physiological challenges, which are significantly affected by dietary factors (nutrients and bioactive compounds). In this review, the results of a series of studies of the potentially beneficial action of ingested PC on some major selected risk factors of ARDs are described. While in vitro and in vivo studies (preclinical) are mainly related to the possible mechanisms of action of PC, RCTs represent a closer approach for the substantiation of the effects of these molecules in humans and allow the possibility to use a health claim for some foods containing them. This relevant information is basic when considering the association of PC with a healthy aging process, since the development of CVD may be retarded through the intake of a myriad of dietary PC affecting the risk factors involved.

Lines 57: change to “Food intake plays a key role in modulation of the risk of cardiometabolic…”

Answer: The phrase was replaced.

New text: Diet plays a major role in the development of CVD.

Rephrase lines 34-36 for clarification

Answer: It was rephrased.

New text: In fact, some dietary priorities have been established in order to promote a general healthy aging, including the intake of minimally processed foods, bioactive rich foods (fruits, nuts, seeds, beans, vegetables, whole grains, plant oils, yogurt, and fish and the reduction of the intake of refined starch, sugars, trans fats and sodium [4]. Accordingly, the current nutritional guidelines for the prevention of CVD include

(Changes are shown in red in version 2)

Reviewer 2 Report

This manuscript reviewed the roles of phenolic compounds on the reduction of risk factors of cardiovascular diseases. Overall, this review provided interesting information about the potential benefits of phenolic compounds on CVD, and the manuscript is well written. Below are several specific comments. 

"ageing" in the manuscript should be revised "aging".

For many abbreviations, please give their full spelling when showing in the first time.

Please add one or two figures to summarize the molecular targets or molecular signaling of phenolic compounds, which can increase the readable of the paper.

Author Response

This manuscript reviewed the roles of phenolic compounds on the reduction of risk factors of cardiovascular diseases. Overall, this review provided interesting information about the potential benefits of phenolic compounds on CVD, and the manuscript is well written. Below are several specific comments. 

- "ageing" in the manuscript should be revised "aging".

Answer: Ageing was replaced by aging in the whole manuscript

- For many abbreviations, please give their full spelling when showing in the first time

Answer: done

Please add one or two figures to summarize the molecular targets or molecular signaling of phenolic compounds, which can increase the readable of the paper.

Answer: no figure was added, however Table 1 was modified and the text was improved in order to increase its readability.

Reviewer 3 Report

The review summarizes the main preclinical and clinical evidence of polyhenols' physiological effecs.

It is generally well written and only few senteces are not so easy to read. I suggest an English style revision.

I suggest to add a Table summarizing for each compound considered in the text the main acion mechanism.

Author Response

The review summarizes the main preclinical and clinical evidence of polyhenols' physiological effects.

It is generally well written and only few sentences are not so easy to read. I suggest an English style revision.

Answer: done. The whole manuscript as revised and many phrases were modified.

I suggest to add a Table summarizing for each compound considered in the text the main action mechanism.

Answer: Table 1 was modified and the correspondent text was improved in order to increase its readability.

(Changes are shown in red in version 2)

Round 2

Reviewer 1 Report

Authors have addressed the majority of my comments and significantly improved the manuscript